# Reverse Osmosis with Intermediate Chemical Demineralization: Scale Inhibitor Selection, Degradation, and Seeded Precipitation

**DOI:** 10.3390/molecules29102163

**Published:** 2024-05-07

**Authors:** Shichang Xu, Ping Wang, Lixin Xie, Yawei Du, Wen Zhang

**Affiliations:** 1Tianjin Key Laboratory of Membrane Science and Desalination Technology, State Key Laboratory of Chemical Engineering, School of Chemical Engineering and Technology, Tianjin University, Tianjin 300350, China; xu_sc1@tju.edu.cn (S.X.); pingwang_971213@tju.edu.cn (P.W.); 2School of Chemical Engineering and Technology, Hebei University of Technology, Tianjin 300401, China; sonicduyawei001@126.com

**Keywords:** RO concentrate, CaSO_4_, scale inhibitor, UV/H_2_O_2_ degradation, seeded precipitation

## Abstract

Two-stage reverse osmosis (RO) processes with intermediate concentrate demineralization (ICD) provide an efficient strategy to treat brines with high CaSO_4_ contents and reduce concentrate discharge. In this paper, an SRO concentrate is treated using ICD to remove CaSO_4_ and then mixed with a PRO concentrate for further desalination in SRO, thereby reducing the discharge of the concentrate. We investigate the selection and degradation of scale inhibitors, as well as seeded precipitation in the two-stage RO process with ICD, to achieve a high water recovery rate. A scale inhibitor is added to restrain CaSO_4_ crystallization on the membrane surface, and the optimized scale inhibitor, RO-400, is found to inhibit calcium sulfate scaling effectively across a wide range of the saturation index of gypsum (SIg) from 2.3 to 6. Under the optimized parameters of 40 W UV light and 70 mg/L H_2_O_2_, UV/H_2_O_2_ can degrade RO-400 completely in 15 min to destroy the scale inhibitor in the SRO concentrate. After scale inhibitor degradation, the SRO concentrate is desaturated by seeded precipitation, and the reaction degree of CaSO_4_ reaches 97.12%, leading to a concentrate with a low SIg (1.07) for cyclic desalination. Three UVD-GSP cycle tests show that the reused gypsum seeds can also ensure the effect of the CaSO_4_ precipitation process. This paper provides a combined UVD-GSP strategy in two-stage RO processes to improve the water recovery rate for CaSO_4_-contained concentrate.

## 1. Introduction

Reverse osmosis (RO) has been widely used in brine desalination because of its simple process and low energy consumption [1,2,3]. After brine treatment, RO produces a large amount of salt-containing concentrate with a complex composition [4,5]. Secondary reverse osmosis (SRO) can further treat the salt-containing concentrate to reduce the amount of high-concentration brine. The two-stage RO process integrating RO desalination with intermediate concentrate demineralization (ICD) is one of the promising strategies to overcome the recovery limitation caused by mineral scaling during RO desalination [6,7,8,9,10,11]. ICD can reduce the concentration of calcium sulfate in the RO concentrate, allowing it to be further desalted in a subsequent SRO process [6,12].

In the concentrated water of primary reverse osmosis (PRO), there is usually a high concentration of calcium sulfate (CaSO_4_). For example, the saturation index of CaSO_4_ (SIg) can reach up to 0.7 in the concentrated water of PRO for the treatment of brine in the coal chemical industry [13]. Because of concentration polarization (CP) [14], the SIg on the RO membrane surface can exceed 4 when the SRO recovery rate is 70% [15]. Hence, a scale inhibitor is necessary before SRO to avoid mineral salt scaling on the surface of the SRO membrane [16,17,18]. After SRO, the scale inhibitors in the SRO concentrate need to be destroyed to precipitate CaSO_4_ [17,19]. Accordingly, we propose the following two-stage RO process with ICD to treat brines (Figure 1), where the SRO concentrate is treated with ICD to remove CaSO_4_ and then mixed with the PRO concentrate for further desalination in SRO.

In the ICD process, gypsum seeded precipitation (GSP) has been used to remove CaSO_4_ [7,20,21]. Before seeded precipitation, a scale inhibitor should be removed because of its inhibiting effect on the crystallization process of CaSO_4_ [17,19]. Coagulation is a conventional method for removing scale inhibitors. For example, polyaluminum chloride and surfactant sodium dodecyl sulfate have been used to adsorb scale inhibitors for the formation of CaSO_4_ precipitation However, the residual product will aggravate the surface fouling of the RO membrane [22,23,24]. Adjusting the pH can induce the precipitation of CaCO_3_ [7,20,21]. However, it is not effective for the CaSO_4_ system [8]. Therefore, it is necessary to develop new methods to remove scale inhibitors before gypsum seeded precipitation.

The degradation of scale inhibitors is a promising method for removing scale inhibitors [25,26,27]. For example, ultraviolet-driven persulfate oxidation (UV/PS) is used to degrade scale inhibitors and promote CaCO_3_ precipitation successfully [19]. However, the use of persulfate as an oxidant will introduce new impurity ions to the RO system and increase the salt content. H_2_O_2_ is also a common oxidant that will not introduce new impurities and salts [28,29]. Nevertheless, there are few studies on UV/H_2_O_2_ in the treatment of RO concentrate with a high concentration of CaSO_4_. In addition, it is important to study the whole process of SRO to treat concentrated solutions containing high calcium sulfate. However, there are few reports about the process.

This paper systematically addresses the optimization of three components of the UVD-GSP process, which include the following: scale inhibitor selection, degradation, and seed precipitation. We investigate the inhibiting effect of different scale inhibitors on the CaSO_4_ concentrate first and then develop an ICD process using UV/H_2_O_2_ degradation of the scale inhibitor (UVD) and gypsum seeded precipitation (GSP) to treat the RO concentrate with a high concentration of CaSO_4_ (Figure 2). Four common commercial-scale inhibitors were used to treat the CaSO_4_ concentrate by static scale inhibition experiments. The effects of scale inhibitor dose, pH, and temperature were studied for the selected inhibitors. Then, the feasibility of removing scale inhibitors from RO concentrates by UVD was studied, and the removal efficiency was explored in relation to UV intensity, H_2_O_2_ dose, and light pretreatment time. In subsequent GSP, the effect of seed size, loading amount, and recyclability on gypsum precipitation performance was assessed and optimized to obtain a high reaction degree of CaSO_4_ and a low saturation index of gypsum. This paper provides a clean and highly efficient method based on UV/H_2_O_2_ to remove scale inhibitors from RO concentrates, without the introduction of impurities into the entire system. The whole process studied to treat the concentrated solution containing high calcium sulfate by SRO, including scale inhibitor selection, degradation, and seed precipitation, can also provide guidance for actual production.

## 2. Results and Discussion

### 2.1. Static Scale Inhibition Experiment

We evaluated the inhibition performance of different scale inhibitors. In Figure 3a, with the gradual increase in the SIg, scaling is observed in the simulated concentrates with different scale inhibitors. The SIg values of TH-1100, SI-001, EDTMPS, and RO-400 are 3, 3.5, 4, and 6, respectively. Therefore, RO-400 has the best scale inhibition effect. Figure 3b is a digital photograph of the model concentrate after adding different scale inhibitors for 10 h when the SIg = 6.5. It can be seen that the amount of scale in the model concentrate with RO-400 is the least. SEM images of CaSO_4_ precipitation are shown in Figure 3c. In the presence of TH-1100, the micro-morphology of CaSO_4_ precipitation is basically the same as that without the scale inhibitor (Appendix A).

In the presence of RO-400, the crystal structure of CaSO_4_ is damaged dramatically. There are only loose and broken small particles, without the original plate-like structure of CaSO_4_. These small particles are easily washed away by backwashing, thereby reducing the adverse effects of CaSO_4_ precipitation. Therefore, RO-400 has the best anti-CaSO_4_ precipitation performance across a wide range of the SIg from 2.3 to 4. This may be due to the fact that the carboxyl group on the main chain of the polymer is an effective functional group to inhibit the formation of scale [30,31]. Therefore, in the following experiments, RO-400 was used as the optimized scale inhibitor.

The effects of dose, pH value, and temperature on scale inhibition performance were also studied for the RO-400 inhibitor. In Figure 3d, η initially increases with the increase in the RO-400 dose and then reaches 100% at the RO-400 dose of 3 ppm [32]. In Figure 3e, the scale inhibitor performs well across a wide range of pH values from 5 to 11. This pH range is basically consistent with that of industrial RO influent, indicating the high industrial application potential of RO-400. In Figure 3f, η is relatively stable in the range of 15–30 °C. However, it should be noted that when the temperature is higher than 30 °C, η may decrease because of the enhanced molecular movement, and a higher dose of RO-400 may be required to cope with this situation [33].

### 2.2. UVD-GSP Processes

The effects of UVD operating parameters on the performance of UVD-GSP were studied here, including UV intensity, H_2_O_2_ dose, and light pretreatment time (*t_p_*) [29]. The results were compared to the contrast groups, the PBL group without RO-400 (RO-400-free), and the NBL group containing 10 ppm RO-400 but without UVD treatment (RO-400 + UVD-free). The PBL and NBL groups represent the upper and lower limits of the scale inhibitor removal effect in the UVD process, respectively.

Figure 4a demonstrates the effect of UV intensity on the UVD-GSP processes. In the UVD stage, when using a UV light of 20 W for the irradiation of 15 min, 3% of the scale inhibitor remained (Appendix A). Then, 2.5 g/L (7–43 µm) of seed crystal is added. After the 180-min GSP desaturation process, the precipitation degree of CaSO_4_ reaches 87.8%, and the SIg is 1.25. The residual scale inhibitor ratio (R) gradually becomes smaller with the increase in UV intensity. When the light intensity reaches 40 W, there is no residual scale inhibitor in the UVD stage. After the 180-min GSP desaturation process, the precipitation degree of CaSO_4_ increases to 97.12%, and the SIg decreases to 1.07. The kinetic curve of CaSO_4_ precipitation in this situation is very close to the curve of GSP without the scale inhibitor.

Figure 4b shows an SEM image of CaSO_4_ precipitation at t = 35 min of the UVD-GSP processes. With the increase in UV intensity in the UVD stage, the crystal structure of CaSO_4_ in the GSP process is more complete, indicating the more significant degradation of the scale inhibitor. At a fixed dose of H_2_O_2_, the increase in UV intensity can increase the concentration of hydroxyl radicals in the concentrate, leading to the improved degradation of scale inhibitors in the UVD process [34].

Figure 4c illustrates the effect of H_2_O_2_ dose on the performance of UVD-GSP. At the UVD stage, 10 mg/L H_2_O_2_ is added to the model concentrate B-bulk. After 15 min of irradiation under a 40 W UV light, 7% of the scale inhibitor remains (Appendix A). Then, 2.5 g/L (7–43 µm) of crystal seeds is added. After the 180-min GSP desaturation process, the precipitation degree of CaSO_4_ reaches 86.24%, and the SIg is reduced to 1.28, which is still far from the curve of PBL. When the H_2_O_2_ dose increases from 10 mg/L to 70 mg/L, the scale inhibitor content is reduced from 7% to 0% after the UVD stage (Appendix A). In the subsequent GSP process, the precipitation degree of CaSO_4_ and the SIg gradually approach the PBL baseline. This is due to the fact that the concentration of hydroxyl radicals in the concentrated solution is determined by the dose of H_2_O_2_. Increasing the dose of H_2_O_2_ will increase the concentration of hydroxyl radicals and enhance the degradation of scale inhibitors.

Figure 4d presents the effect of light pretreatment time (*t_p_*) on the UVD-GSP processes. In the UVD stage, 70 mg/L H_2_O_2_ is added, and 3% of the scale inhibitor remains after 15 min of irradiation with a 20 W UV light (Appendix A). Subsequently, after the 180-min GSP desaturation process (adding 2.5 g/L, 7–43 µm seeds), the precipitation degree of CaSO_4_ only reaches 87.8% and the SIg is 1.25, which deviates from the baseline PBL seriously. The residual scale inhibitor is completely removed by extending the UVD pretreatment time from 15 min to 20 min, where the precipitation degree of CaSO_4_ reaches 97.12%, and the SIg is reduced to 1.07. This indicates that the oxidative degradation of scale inhibitors by hydroxyl radicals in UVD requires a certain amount of time, and prolonged UV light treatment time can increase the extent of scale inhibitor degradation, thus reducing the inhibitory effect in the GSP process [34].

As shown above in Figure 4a, in the model concentrate B-bulk with 70 mg/L H_2_O_2_, 3% of the scale inhibitor remains after 20 W UV irradiation for 15 min (Appendix A). We fixed this UVD operating condition and explored the effects of seed particle size and loading amount on the GSP process.

In Figure 5a, after the 180-min GSP process, with the increase in seed particle size, the precipitation degree of CaSO_4_ decreases from 87.8% to 77.98%, and the SIg increases from 1.25 to 1.43. That is, under the same seed loading, the larger the seed particle size, the worse the desaturation of CaSO_4_. This is because the seed particles with a larger size provide a smaller active surface for slower CaSO_4_ precipitation [35].

In Figure 5b, when the seed loading increases from 1.5 g/L to 5.5 g/L (7–43 µm), the precipitation degree of CaSO_4_ increases from 70.86% to 94.58%, and the SIg decreases from 1.56 to 1.12 after the 180-min GSP process. This is because the increase in seed loading can provide a more active surface for the growth of CaSO_4_ crystals, overcoming the blocking effect of residual scale inhibitors and accelerating the kinetic process for the formation of CaSO_4_ precipitation [35].

In addition, we also investigated the reusability of the seeds. The parameters of 2.5 g/L seeds with a size of 7–43 μm are used for the GSP cycle experiments. In order to eliminate the adverse effects of residual scale inhibitors, GSP cycle experiments were carried out after UVD without scale inhibitor residues (40 W, 70 mg/L, 15 min). In Figure 5c, after adding the initial seeds, the kinetic curve is almost coincident with the PBL baseline. In the first cycle, the kinetic curve has a slight deviation from the PBL baseline. In the third cycle, the precipitation degree of CaSO_4_ in the GSP process is 91.53%, and the SIg reaches 1.18, suggesting the seed cycle performance is satisfactory.

Figure 5d shows the particle size distribution of the recycled seeds. It can be seen that the particle size distribution of the seeds gradually migrates to larger sizes after the cycle experiments. In Figure 5e, after three cycles of GSP experiments, the surface of the seeds is still smooth, and the morphology of the seeds has not changed much from the initial seeds. However, the seed particle size becomes larger after recycling.

In summary, during the UVD process, the optimal parameters are a 40 W ultraviolet lamp and 70 mg/L H_2_O_2_ with 15 min illumination, and there is no residual scale inhibitor under this operating condition. However, taking cost into account, the best UVD operating conditions are a 40 W UV lamp and 40 mg/L H_2_O_2_ with 15 min illumination. Under the above operating conditions, although there is a trace amount of scale inhibitor residue (1%), the precipitation degree of CaSO_4_ (95.43%) and the SIg (1.10) after the 180-min GSP process are generally close to the PBL baseline.

## 3. Experiment

### 3.1. Materials and Solutions

#### 3.1.1. Materials

NaCl (99.5%) and CaSO_4_·2H_2_O (98%) were acquired from Tianjin Heowns Biochemical Technology Co., Ltd. Na_2_SO_4_ (99%), CaCl_2_ (96%), H_2_O_2_ (30%), HCl (36%-38%), and NaOH (96%) were purchased from Tianjin Jiang Tian Chemical Technology Co., Ltd. Three different particle sizes of CaSO_4_·2H_2_O crystal seeds, 7–43 µm, 43–71 µm, and 71–91 µm, were obtained by sieving. All the reagents were used directly without further purification. Solutions were prepared using deionized water.

#### 3.1.2. Scale Inhibitors

Currently, the scale inhibitors widely used in RO mainly include two types, one is phosphonate-based and the other is acrylic acid-based [30]. The commercial scale inhibitors used in this study consisted of two organophosphines and two polymers, 2-phosphonobutane-1,2,4-tricarboxylic acid (SI-001, Polymer Technology) (Figure 6a), ethylene diamine tetra methylene phosphonic acid sodium (EDTMPS, Shandong Taihe Technologies Co., Ltd.) (Figure 6b), low-molecular-weight polyacrylic acid partially neutralized salts (TH-1100, Shandong Taihe Technologies Co., Ltd.) (Figure 6c), and low-molecular-weight polyacrylic acid homopolymer (RO-400, BASF SE) (Figure 6d).

#### 3.1.3. Preparation of RO Concentrate

The composition of the PRO concentrate model solution (A) is shown in Table 1. ESNA1-LF2-LD of FilmTec RO membrane modules from DOW were included in the case studies. The software IMSDesign (Version 2.229.87%) was used to calculate the bulk concentration (solution B-bulk) and the membrane surface concentration (solution B-membrane surface) when the SRO water recovery was 70% [15]. The specific composition is shown in Table 1.

### 3.2. Experimental Details

#### 3.2.1. Static Scale Inhibition Experiments

Static scale inhibition experiments were carried out according to GB/T 16632-2019 [36,37]. Specifically, simulated concentrated solutions of different SIg values were prepared, and the compositions are shown in Appendix A. First, 8 ppm of TH-1100, SI-001, EDTMPS, or RO-400 was added to the simulated concentrated solution, stirred at a constant temperature for 10 h, and then filtered with a 0.22 µm disposable filter. The content of Ca^2+^ in the filtrate was determined by EDTA complexometric titration [38]. After selecting the appropriate scale inhibitor, the operating conditions including scale inhibitor dose, pH, and temperature were optimized using the model concentrate solution of B-membrane surface (Table 1).

The effectiveness of the scale inhibitor was evaluated by the scale inhibition efficiency (*η*, %), which was calculated according to the following equation,
(1)$$\eta =\frac{c_{2}-c_{1}}{{c_{0}-c}_{1}}$$
where $c_{0}$ (mmol/L) represents the initial concentration of Ca^2+^, $c_{1}$ (mmol/L) indicates the concentration of Ca^2+^ in the filtrate of the samples without the scale inhibitor, and $c_{2}$ (mmol/L) stands for the concentration of Ca^2+^ in the filtrate of the samples with the scale inhibitor.

#### 3.2.2. UVD-GSP Experiments

ICD consists of two consecutive steps, UVD and GSP. For UVD, a certain amount of H_2_O_2_ was added to the prepared 1 L model concentrate B-bulk (10 ppm of RO-400) and irradiated under ultraviolet light for a certain time. The effects of light intensity, H_2_O_2_ dose, and light pretreatment time (*t_p_*) on the UVD stage were investigated. Then, gypsum seeds were added to the model concentrate B-bulk after UV irradiation to realize the GSP process. The effects of seed particle size, loading amount, and recyclability of the seeds on GSP were explored.

During the experiments for UVD, 20 mL of samples were taken from the reactor at intervals of 5 min to measure the scale inhibitor content. The concentration of RO-400 in water samples was determined according to the turbidity method [39]. The absorbance was measured at a 420 nm wavelength, and the concentration of RO-400 was calculated according to the standard curve (Appendix A). The residual scale inhibitor ratio (*R*) was calculated according to the following equation,
(2)$$R=\frac{c_{As,0}-c_{As}}{c_{As}}$$
where $c_{As,0}$ (ppm) and $c_{As}$ (ppm) refer to the initial and sampling concentrations of the scale inhibitor in model concentrate B, respectively.

During the experiments for GSP, 3 mL of samples were taken from the reactor every 20 min, and the content of Ca^2+^ in the filtrate was measured after filtration. The pH change in solution B-bulk was monitored by pH meter. The saturation index of gypsum (*SIg*) was calculated by the software IMS Design [8]. The equation was as follows,
(3)SIg=IAPKsp, gypsum
where *IAP* is the product of ion activity and $K_{sp,\;gypsum}$ is the solubility product of calcium sulfate.

The reaction degree of CaSO_4_ ($\alpha_{gypsum}$) was determined by the following formula [21],
(4)$$\alpha_{gypsum}=\frac{c_{0}-c}{c_{0}-c_{eq}}$$
where $c_{eq}$ represents the concentration of Ca^2+^ in solution B when CaSO_4_·2H_2_O is in equilibrium (Appendix A).

The morphology of gypsum seeds was characterized by scanning electron microscopy (SEM, Regulus 8100) at 3.0 kV. The particle size distribution of the crystalline seeds used in the cycling experiments was determined by employing shock sieving and weighed.

## 4. Conclusions

To improve the water recovery rate, a two-stage RO process with ICD is proposed to treat RO concentrates with a high CaSO_4_ content. To avoid the scaling of the membrane surface during the SRO operation, scale inhibitors are added into the concentrates. After SRO, the scale inhibitor is destroyed by UV/H_2_O_2_ degradation, and the resulting concentrate can be treated by the gypsum seeded precipitation (GSP) process.

For concentrates containing high CaSO_4_, suitable scale inhibitors were selected from the following three aspects: Ca^2+^ concentration in the concentrate, macroscopic scaling amount, and microscopic CaSO_4_ precipitation morphology. RO-400, being the preferred scale inhibitor, shows an effective inhibition effect across a wide range of SIg (2.3–6) and pH (5–11) values. In the UVD process, a 40 W UV light with 70 mg/L H_2_O_2_ can degrade RO-400 completely in 15 min. In the GSP process, 2.5 g/L seeds with a size of 7–43 µm can decrease the SIg value to 1.07 (approaching the thermodynamic limit of 1) and remove 97% CaSO_4_. Gypsum seeds for the GSP process can be reused three times when there is no scale inhibitor residue in the UVD stage. This paper provides a clean and highly efficient method based on UV/H_2_O_2_ to remove scale inhibitors from RO concentrates without the introduction of impurities to the entire system. The whole process studied to treat the concentrated solution containing high calcium sulfate by SRO, including scale inhibitor selection, degradation, and seed precipitation, can also provide guidance for actual production. Future work will focus on the applicability of the UVD-GSP-based RO treatment for water recovery and concentrate minimization in the desalination of CaSO_4_ brines.

## Figures and Tables

**Figure 1 molecules-29-02163-f001:**
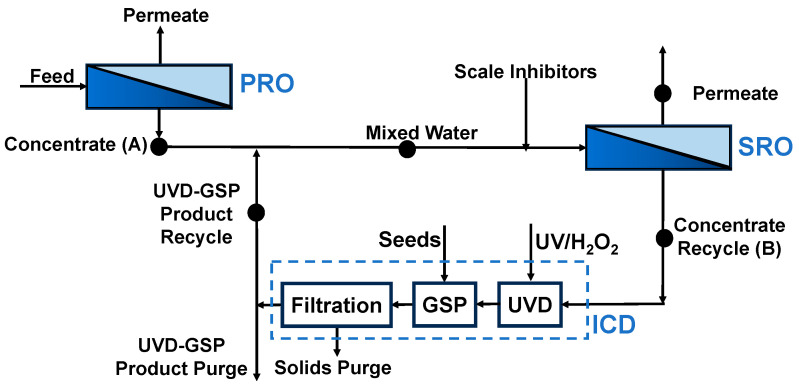
Schematic diagram of the two-stage RO process with ICD (UVD-GSP).

**Figure 2 molecules-29-02163-f002:**
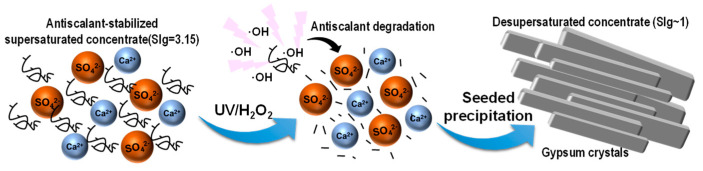
UVD-GSP mechanism for the removal of calcium sulfate from RO concentrates.

**Figure 3 molecules-29-02163-f003:**
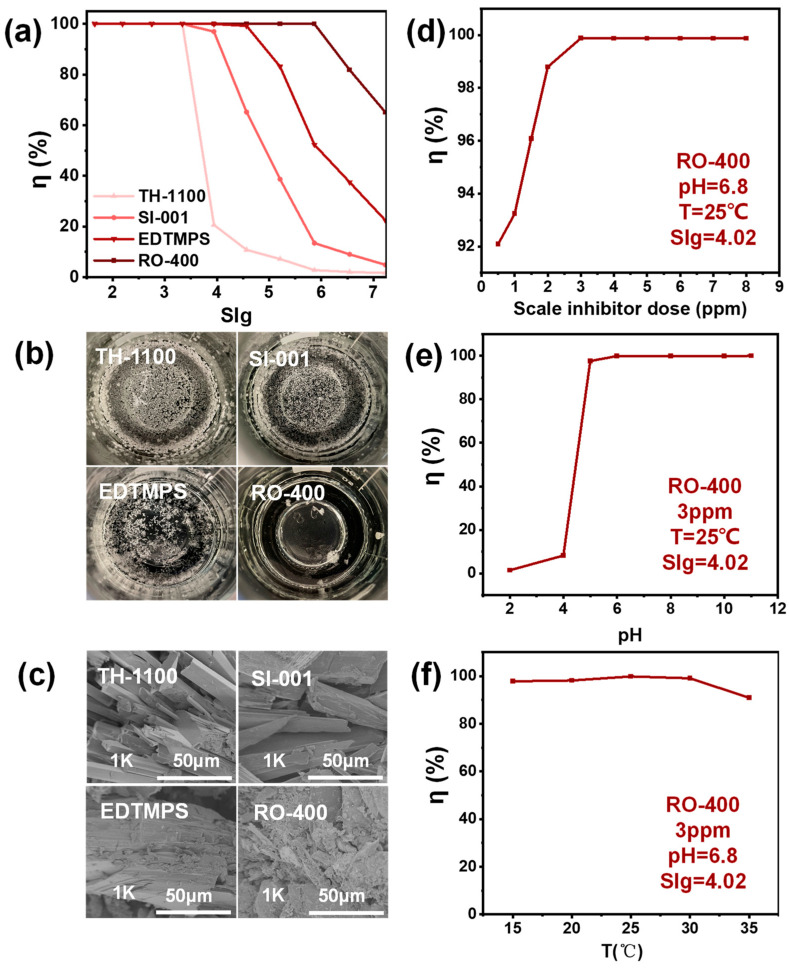
(**a**) The inhibition performance of different scale inhibitors in simulated concentrates; (**b**) digital photographs; and (**c**) SEM images of scaling materials after 10 h with different scale inhibitors (SIg = 6.5). The effect of (**d**) dose, (**e**) pH, and (**f**) temperature on the inhibition performance of RO-400.

**Figure 4 molecules-29-02163-f004:**
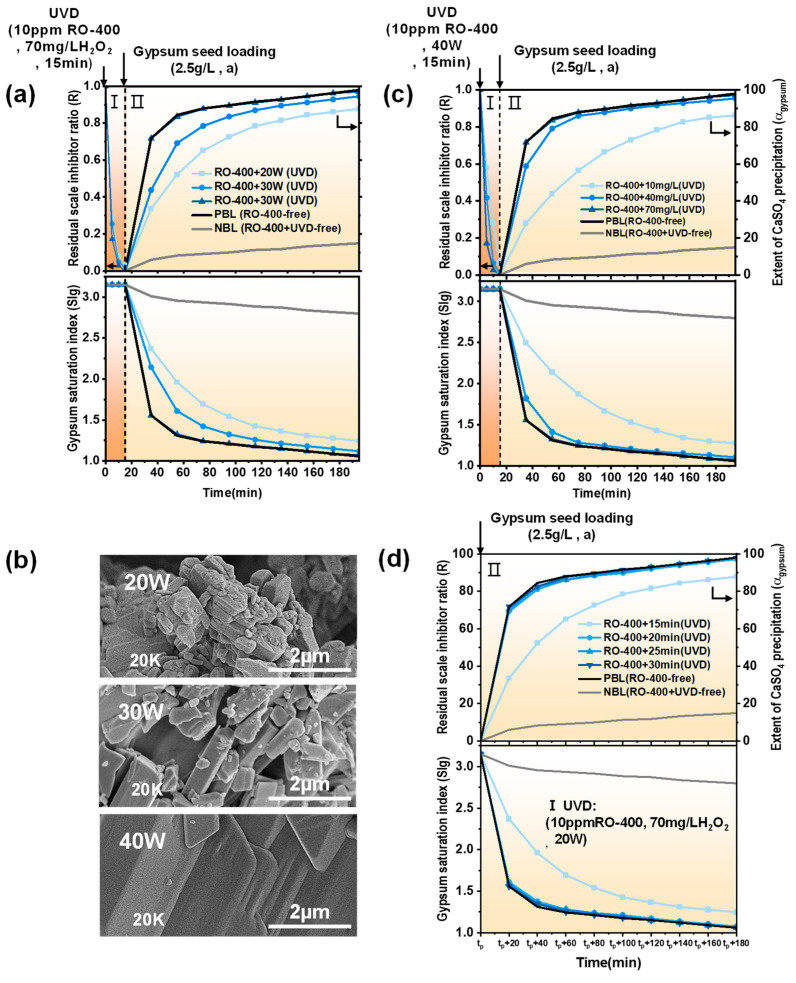
The effect of (**a**) UV intensity, (**c**) H_2_O_2_ dose, and (**d**) *t_p_* on the residual scale inhibitor ratio (*R*) in UVD. (**b**) SEM images of CaSO_4_ in the subsequent GSP process using different UV intensities in UVD (t = 35 min).

**Figure 5 molecules-29-02163-f005:**
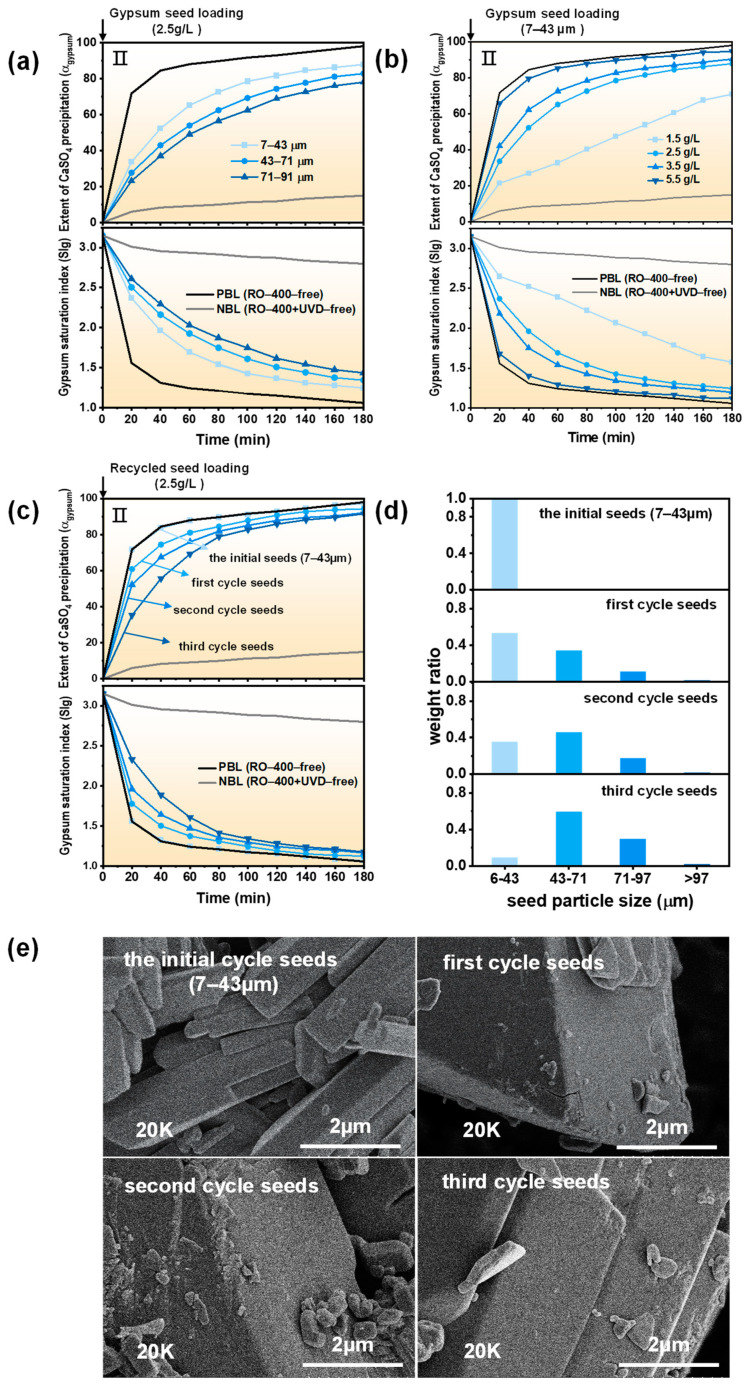
The effect of (**a**) seed particle size, (**b**) loading amount, and (**c**) recyclability of the seeds on the GSP process. (**d**) Particle size distribution and (**e**) SEM images of crystalline seeds added in each set of GSP for seed cycling experiments.

**Figure 6 molecules-29-02163-f006:**
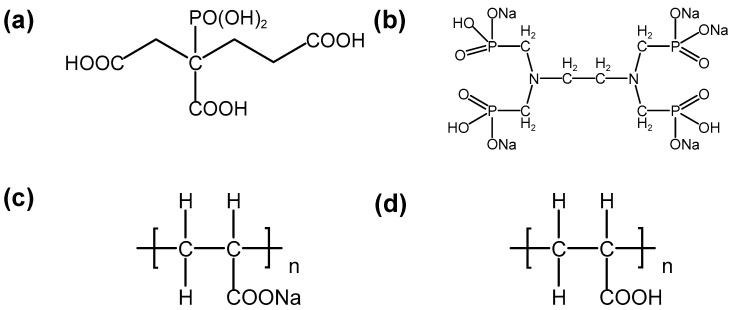
Chemical structure of the main components including (**a**) SI-001, (**b**) EDTMPS, (**c**) TH-1100, and (**d**) RO-400.

**Table 1 molecules-29-02163-t001:** SRO concentrate model solutions for a coal chemical PRO concentrate at a 70% recovery rate.

Salt		PRO Concentrate Model Solution (A)	SRO Concentrate Model Solutions (70%)	
		Composition	Solution (B-Bulk)	Solution (B-Membrane Surface)
CaCl_2_	mmol/L	12.85	42.91	52.22
Na_2_SO_4_	mmol/L	20.75	69.31	84.34
NaCl	mmol/L	26.40	88.20	107.33
pH	-	6.7	6.8	6.9
SIg	-	0.70	3.15	4.02

## Data Availability

Data are contained within the article and Appendix A.

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
