# Peer review of "Reverse Osmosis with Intermediate Chemical Demineralization: Scale Inhibitor Selection, Degradation, and Seeded Precipitation"

_molecules, 2024, doi:10.3390/molecules29102163_

Round 1

Reviewer 1 Report

Comments and Suggestions for Authors

The research paper entitled Reverse Osmosis with Intermediate Chemical Demineralization: Scale Inhibitor Selection, Degradation and Seeded Precipitation, is interesting in general. This paper investigates the effectiveness of different scale inhibitors on CaSO4 concentrate. It then develops an Integrated Chemical Degradation (ICD) process, using UV/H2O2 degradation (UVD) and gypsum seeded precipitation (GSP), for treating RO concentrate with high CaSO4 concentration. It furthermore examines four common scale inhibitors through static inhibition experiments, studying factors like dosage, pH, and temperature. It also explores the feasibility of removing scale inhibitors from RO concentrates via UVD, analyzing UV intensity, H2O2 dosage, and light pretreatment time. Additionally, GSP is assessed and optimized for seed size, loading amount, and recyclability to enhance gypsum precipitation performance.

Some comments deriving from studying the paper are as follows:

-          The Introduction in Section 1 seems to require further elucidation so as to effectively showcase the innovative aspects of this research and underscore its focus on addressing an existing gap in the field under investigation. As per the reviewer's opinion, a more in-depth review in the field would significantly add to the quality of the paper. Moreover, for the benefit of the readership, it is suggested to include a paragraph describing the structure of the article in the end of the introduction section.

-          The methodology employed in this study for experimental purposes, is clearly outlined in Section 2, yielding encouraging results as presented in Section 3, that may inspire other researchers to replicate certain elements of this approach. However, Section 3 is suggested to be renamed as Results and Discussion for presentation purposes. In addition, images in Figures 4, 5 and 6 need to be improved since they appear to contain too much information that is difficult for the reader to distinguish.

-          The conclusions section appears to be relatively concise. To enhance the paper's quality, it would be beneficial to expand this section, delving deeper into the findings deriving by the results of the research as well as and their broader implications. Adding more information about the directions for future research would add to the quality of the paper.

-          Finally, the paper is well-structured in general and written in appropriate English language according to the standards of the Journal, however some minor spell-checking might be of need.

Comments on the Quality of English Language

Minor spell-checking might be of need.

Author Response

Hello reviewer:

Thank you very much for your suggestion, I have placed a response to your suggestion in the attached file for your review.

Reviewer 2 Report

Comments and Suggestions for Authors

A two-stage RO process with scale inhibitor removal is used to treat synthetic RO concentrate with a high CaSO4 content. Key operation parameters were deeply studied. Before the manuscript is accepted, a revision is needed, and some comments are in the following.

How to measure 𝑐𝐴s,0 (ppm) and 𝑐𝐴s (ppm) refer to the initial and sampling concentrations of scale inhibitor?

Line 215, “When the light intensity reaches 40 W, there is no residual scale inhibitor in the UVD stage.” How do you know that? 

The words on Figs.4, 5 and 6 are unclear, including horizontal and vertical coordinateslegends. No magnification factor in the SEM photos.

In Fig. 6, no a, b, c, d, e.

Sig or Sig?

Lines 263-264, “In Figure 6b, when the seeds loading increases from 1.5 g/L to 5.5 g/L (7-43 µm), the precipitation degree of CaSO4 increases from 70.86% to 94.58% and SIg decreases from 1.56 to 1.12 after the 180 min’s GSP process” the seeds loading increased 2.7 times, but Sig only decreased from 1.56 to 1.12, what is the optimal seeds dosage?

What is the optimal operation parameters in this study?

Author Response

(The authors gave the same response as above.)
